# Differences in salivary microbiome among children with tonsillar hypertrophy and/or adenoid hypertrophy

Ying Xu,[1] Min Yu,[1] Xin Huang,[2] Guixiang Wang,[3] Hua Wang,[3] Fengzhen Zhang,[3] Jie Zhang,[3] Xuemei Gao[1]

**ABSTRACT** Children diagnosed with severe tonsillar hypertrophy display discernible craniofacial features distinct from those with adenoid hypertrophy, prompting illuminating considerations regarding microbiota regulation in this non-inflammatory condition. The present study aimed to characterize the salivary microbial profile in children with tonsillar hypertrophy and explore the potential functionality therein. A total of 112 children, with a mean age of $7.79 \pm 2.41$ years, were enrolled and divided into the tonsillar hypertrophy (TH) group ($n = 46$, $8.4 \pm 2.5$ years old), adenoid hypertrophy (AH) group ($n = 21$, $7.6 \pm 2.8$ years old), adenotonsillar hypertrophy (ATH) group ($n = 23$, $7.2 \pm 2.1$ years old), and control group ($n = 22$, $8.6 \pm 2.1$ years old). Unstimulated saliva samples were collected, and microbial profiles were analyzed by 16S rRNA sequencing of V3–V4 regions. Diversity and composition of salivary microbiome and the correlation with parameters of overnight polysomnography and complete blood count were investigated. As a result, children with tonsillar hypertrophy had significantly higher α-diversity indices ($P < 0.05$). β-diversity based on Bray–Curtis distance revealed that the salivary microbiome of the tonsillar hypertrophy group had a slight separation from the other three groups ($P < 0.05$). The linear discriminant analysis effect size (LEfSe) analysis indicated that *Gemella* was most closely related to tonsillar hypertrophy, and higher abundance of *Gemella*, *Parvimonas*, *Dialister*, and *Lactobacillus* may reflect an active state of immune regulation. Meanwhile, children with different degrees of tonsillar hypertrophy shared similar salivary microbiome diversity. This study demonstrated that the salivary microbiome in pediatric tonsillar hypertrophy patients had different signatures, highlighting that the site of upper airway obstruction primarily influences the salivary microbiome rather than hypertrophy severity.

**IMPORTANCE** Tonsillar hypertrophy is the most frequent cause of upper airway obstruction and one of the primary risk factors for pediatric obstructive sleep apnea (OSA). Studies have discovered that children with isolated tonsillar hypertrophy exhibit different craniofacial morphology features compared with those with isolated adenoid hypertrophy or adenotonsillar hypertrophy. Furthermore, characteristic salivary microbiota from children with OSA compared with healthy children has been identified in our previous research. However, few studies provided insight into the relationship between the different sites of upper airway obstruction resulting from the enlargement of pharyngeal lymphoid tissue at different sites and the alterations in the microbiome. Here, to investigate the differences in the salivary microbiome of children with tonsillar hypertrophy and/or adenoid hypertrophy, we conducted a cross-sectional study and depicted the unique microbiome profile of pediatric tonsillar hypertrophy, which was mainly characterized by a significantly higher abundance of genera belonging to phyla *Firmicutes* and certain bacteria involving in the immune response in tonsillar hypertrophy, offering novel perspectives for future related research.

**KEYWORDS** tonsillar hypertrophy, oral microbiome, 16s rRNA sequencing

Address correspondence to Jie Zhang, stzhangj@263.net, or Xuemei Gao, xmgao@263.net.

Jie Zhang and Xuemei Gao contributed equally to this article.

The authors declare no conflict of interest.

See the funding table on p. 15.

10.1128/msystems.00968-24 **1**

The tonsils and adenoids are components of the Waldeyer's lymphatic ring. Tonsils, or palatine tonsils, are located between the anterior and posterior tonsillar pillars; adenoids, also known as pharyngeal tonsils, are located in the nasopharynx (1). It is generally believed that the tonsils usually increase in size throughout childhood and tend to regress during adolescence (2). The incidence of palatine tonsillar hypertrophy (generalized as tonsillar hypertrophy here) in primary school children is 11% (3), and it stands as one of the primary risk factors for pediatric obstructive sleep apnea (OSA), characterized by recurrent partial or complete collapse and obstruction of the upper airway during sleep (4). It can lead to a spectrum of serious consequences, including learning and behavioral difficulties, as well as complications affecting cardiovascular, metabolic, endocrine systems, and growth disorders (5). Recognizing its potential impact on children's health, the management of tonsillar hypertrophy becomes paramount.

Although both tonsillar and adenoid hypertrophy can result in mouth breathing and snoring, numerous studies have found that they exert different consequences on abnormal growth and development of the dentomaxillofacial morphology (6–9). Tonsillar hypertrophy, due to blocking the lower segment of the upper airway, tends to prompt children to advance the mandible forward to increase the width of the oropharyngeal airway, predisposing them to anterior crossbite (8, 10). Besides, our previous research has discovered that children with adenotonsillar hypertrophy and those with isolated adenoid hypertrophy exhibit similar craniofacial morphology and structure, but they differ from children with isolated tonsillar hypertrophy (11). The distinct clinical manifestations of tonsillar hypertrophy, and the fact that pharyngeal lymphoid tissue can enlarge at different sites that cause upper airway obstruction at different levels, sparked our interest in further specialized research regarding the underlying biological mechanisms.

Given that the tonsils serve as a lymphoid organ located at the shared entry of both the gastrointestinal and the respiratory tract (12, 13), studies have explored the microbial interactions in the context of tonsillar diseases (14, 15), including chronic tonsillitis (CT) and tonsillar hypertrophy (TH) (16). Chronic tonsillitis is typically considered as the persistent and recurrent inflammation of the tonsils, characterized by infection (17), whereas tonsillar hypertrophy is regarded as a result of parenchymal hyperplasia or fibrinoid degeneration without symptoms of infection (18, 19). Innate immune and inflammatory responses are found to be more active in simple hypertrophic tonsils, rather than hypertrophic tonsils with recurrent inflammation, and tonsillar hypertrophy might be regulated by diverse immune mechanism (20). However, few studies have investigated the microbiome of pediatric patients with tonsillar hypertrophy, and tonsillar hypertrophy has often been used as a control group for chronic tonsillitis in past studies, with most of them relying on bacterial culture, which could only reflect a small fraction of the entire microbial community (21–24), leaving a knowledge gap concerning the microbiome profile of tonsillar hypertrophy comparing with adenoid hypertrophy.

16S rRNA sequencing analysis of the oral microbiome, which overcomes the limitations of bacterial culture, has emerged as a novel research method in recent years, and characteristic salivary microbiota has been discussed in children with OSA (25). Previous studies have reported the association between gut microbiota and pediatric OSA (26), with their regulatory relationship being explored (27, 28), which further underscores the importance of investigating the relationship between tonsillar hypertrophy and the microbiota. Collecting saliva samples is non-invasive and far more convenient (29), and relatively stable over time (30), thus specially suitable for infants and children (31). Moreover, existing studies have indicated that saliva and the tonsillar surface share a significant portion of their microbial composition (32, 33). Besides, many studies have shown that the oral microbiome plays an important role in maintaining oral and systemic health (34). Investigating the salivary microbiome holds promise for enhancing our comprehension of the characteristics of tonsillar hypertrophy and facilitating the disclosure of the potential relationship between tonsillar hypertrophy and oral microbiota, as well as other physiological factors (35).

Therefore, the present study focuses on pediatric tonsillar hypertrophy and aims to delineate their salivary microbiome profile in comparison with different sites of upper airway obstruction and different degrees of tonsillar hypertrophy based on the evaluation of the V3–V4 region of the 16S rRNA genes. The exploration is expected to contribute to a deeper understanding of the characteristics of tonsillar hypertrophy, exploring the underlying impact of the salivary microbiome, particularly the potential role of microbiota in immune mechanisms, and providing new insights for the early diagnosis of pediatric tonsillar hypertrophy.

## MATERIALS AND METHODS

### Study population

This study used Micropower package (36), a simulation-based method for PERMA-NOVA-based β-diversity comparisons, to simulate distance matrices based on the oral microbiome database from the Human Microbiome Project (HMP) (37) to assess the effect size and statistical power, according to a previous study on the salivary microbiome of children with OSA (25). The effect sizes ($\omega$) (2) were calculated using the simulated matrixes of 80% and 90% powers for varying sample numbers per group (Table S1). An effect size of $\omega^2 = 0.027$ for 20 subjects per group was found to be smaller than those in published microbiome studies of antibiotic exposure analyzed using unweighted Jaccard distances. Therefore, 20 subjects per group are believed to provide adequate statistical power for the primary outcome measure.

The hypertrophy groups were pediatric patients aged 3–14 years, recruited from the Department of Otolaryngology, Head and Neck Surgery and the Department of Sleep Center of Beijing Children's Hospital, Capital Medical University, from February 2023 to October 2023. The isolated tonsillar hypertrophy (TH) group, consisting of a larger tonsillar hypertrophy group with 26 children diagnosed with Grade III tonsillar hypertrophy and a smaller tonsillar hypertrophy group with 20 children diagnosed with Grade II tonsillar hypertrophy through specialist examination; the adenoid hypertrophy (AH) group, comprising 21 children diagnosed with Grade III or IV adenoid hypertrophy through fibro-laryngoscopic examination (38); and the adenotonsillar hypertrophy (ATH) group, which included 23 children diagnosed with Grade III tonsillar hypertrophy and Grade III or IV adenoid hypertrophy simultaneously.

The control group of 22 children with age matched to the hypertrophy group were recruited from children who were undergoing their pre-orthodontic examination at Peking University School and Hospital of Stomatology with the lateral cephalometric radiographs showing no tonsillar or adenoid hypertrophy.

The exclusion criteria for all groups were (1) intake or injection of antibiotics and anti-inflammatory drugs within the last 3 months; (2) history of acute tonsillitis; (3) suffering from severe oral diseases: dental caries (DFMT ≥4), retained roots or crowns, presence of abscess or fistula in the oral cavity, periodontal disease, oral mucosal lesions, oral tumors, oral cancer; (4) presence of systemic diseases, including cardiovascular, cerebrovascular, metabolic, neurologic, respiratory system, hematologic, gastrointestinal, renal, genital–urinary, or thyroid diseases, diabetes, autoimmune diseases, immunodeficiency, congenital diseases, genetic diseases, other chronic inflammatory disease known from the anamnestic interview of the patients, or severe skeletal deformities; (5) active infection (bacterial, fungal, or viral); (6) presence of sleep disorders other than OSA; (7) receipt of the following treatments within the past four weeks: dental procedures, use of mouthwashes, use of immunosuppressants, or intake of medications to regulate the microbiota; (8) with a particular diet; and (9) the presence of pets in the home.

### Fibro-laryngoscopic examination, cephalometric analysis, and clinical examination

The examination on tonsils and adenoids of children from hypertrophy groups was conducted by otolaryngologists in the outpatient clinic. Tonsillar hypertrophy was

evaluated according to the three-grade scale system (39). The degree of adenoid hypertrophy was documented as the percentage obstruction of the measured distance between the anterior and posterior surfaces of the nasopharynx and evaluated using the 1–4 grading scale through fibro-laryngoscopic examination (40).

The examination on tonsils and adenoids of children from the control group was conducted on pre-orthodontic cephalometric examinations performed by radiology specialists using orthopantomograph OC200 digital X-ray machine (Instrumentarium Dental, Tuusula, Finland). The lateral cephalograms were taken with children in an upright position and the Frankfort horizontal parallel to the floor. All children were instructed to remain still and maintain centric occlusion without moving heads or making speech or swallowing. Cephalometric analysis was performed by a single investigator.

The oral examination of all the participants was performed by one qualified dentist. The caries experience was evaluated using the decayed, missing, and filled tooth (dmft/DMFT) index, according to the criteria proposed by World Health Organization (2013) (41).

## Overnight polysomnography (PSG)

PSG examinations were conducted in the sleep center for children in hypertrophy groups before the otolaryngology surgery using a Compumedics E-series PSG System (Compumedics, Australia) or Alice 5Diagnostic Sleep System (Respironics, USA). PSG was interpreted by two technicians and one pediatrician trained in sleep medicine who were blinded to the clinical findings. Respiratory events were scored as obstructive apnea or hypopnea according to the criteria established by American Academy of Sleep Medicine (AASM) (42). The main recorded monitoring indices included obstructive apnea–hypopnea index (OAHI) and lowest oxygen saturation ($SaO_2$). OAHI was calculated as the average number of obstructive apneas and hypopneas per hour of sleep, and the diagnosis of OSA was confirmed by OAHI ≥1 (43).

## Complete blood count (CBC)

Blood samples were routinely collected for complete blood count (CBC) from children in the hypertrophy groups who were scheduled for tonsillectomy, adenoidectomy, or adenotonsillectomy on the day of admission to the surgical ward of the Department of Otolaryngology, Head, and Neck Surgery. Blood samples were drawn from the cubital vein using BD Vacutainer and vacuum tube needles according to standard operating procedures. CBC was measured by automated hematological analyzer Sysmex XS-800i. The analyzer was maintained according to the manufacturer's instruction. The analyzed CBC parameters included white blood cell count (WBC), red blood cell count (RBC), hemoglobin (Hb), hematocrit (HCT), mean corpuscular volume (MCV), mean corpuscular hemoglobin (MCH), mean corpuscular hemoglobin concentration (MCHC), red blood cell distribution width (RDW), platelet count (PLT), mean platelet volume (MPV), platelet distribution width (PDW), plateletcrit (PCT), large platelet ratio (LPR), neutrophil percentage (N), absolute neutrophil count (ANC), lymphocyte percentage (L), absolute lymphocyte count (ALC), monocyte percentage (M), absolute monocyte count (AMC), eosinophil percentage (E), absolute eosinophil count (AEC), basophil percentage (B), and absolute basophil count (ABC). These 23 parameters were categorized as red blood cell-related parameters (RBC, Hb, HCT, MCV, MCH, MCHC, and RDW), white blood cell-related parameters (WBC, N, ANC, L, ALC, M, AMC, E, AEC, B, and ABC), and platelet-related parameters (PLT, MPV, PCT, PDW, and LPR).

## Saliva sample collection

The participants were required to collect 1.5–2 mL of unstimulated saliva after waking up in the morning and a fast of at least 6 h, before oral hygiene procedures and between 6:00 am and 8:00 am. The saliva samples of the hypertrophy and control groups

were collected at the Sleep Center of Beijing Children's Hospital and Department of Orthodontic, Peking University School and Hospital of Stomatology, respectively. The non-stimulated saliva collection process supervised by a dentist took within 15 min. All participants were asked to sit in a comfortable position in a quiet environment, with their heads naturally tilted downwards, and then were instructed to drool or split saliva directly into the labeled 50-mL collection tube (Falcon, sterile conical polypropylene tube with flat-top screw cap). After collection, the samples were immediately placed in an ice box and transferred to the laboratory for processing within 1 h. Pellet bacteria by centrifugation at 10,000×$g$ for 10 min at 4℃. The pellets were stored at −80℃ before DNA extraction (25).

## DNA extraction and 16S rRNA gene sequencing

The total bacterial DNA was extracted using the QIAamp DNA mini kit (Qiagen, Hilden, Germany). The quantity and quality of extracted DNAs were measured using a NanoDrop NC2000 spectrophotometer (Thermo Fisher Scientific, Waltham, MA, USA) and agarose gel electrophoresis, respectively. The V3–V4 hypervariable region of the 16S rRNA gene was PCR amplified. Sample-specific 7-bp barcodes were incorporated into the primers for multiplex sequencing. The PCR components contained 5 µL of buffer (5×), 0.25 µL of Fast pfu DNA Polymerase (5 U/µl), 2 µL (2.5 mM) of dNTPs, 1 µL (10 uM) of each forward and reverse primer, 1 µL of DNA template, and 14.75 µL of ddH2O. Thermal cycling consisted of initial denaturation at 98℃ for 5 min, followed by 25 cycles consisting of denaturation at 98℃ for 30 s, annealing at 53℃ for 30 s, and extension at 72℃ for 45 s, with a final extension of 5 min at 72℃. The PCR amplicons were purified with Vazyme VAHTSTM DNA Clean Beads (Vazyme, Nanjing, China) and quantified using the Quant-iT PicoGreen dsDNA Assay Kit (Invitrogen, Carlsbad, CA, USA). After the individual quantification step, amplicons were pooled in equal amounts, and pair-end 2 × 250 bp sequencing was performed using the Illumina NovaSeq platform with NovaSeq 6000 SP Reagent Kit (500 cycles) at Shanghai Personal Biotechnology Co., Ltd (Shanghai, China).

## Bioinformatics and statical analysis

In this study, the sequencing analysis was performed with the QIIME2 (44) and R package (v3.2.0). Divisive Amplicon Denoising Algorithm 2 (DADA2) (45) was used to quality filter, denoise, merge, and remove chimeras from raw sequence data and to generate amplicon sequence variants (ASVs). Taxonomy was assigned to ASVs using a scikit-learn naive Bayes taxonomy classifier in feature-classifier plugin (46) against the Human Oral Microbiome Database (HOMD) (47). Alpha diversity measures the diversity of microorganisms within a sample. It provides insights into the variety and abundance of different microorganisms present in a sample. ASV-level alpha-diversity indices, such as Chao1 richness estimator, Observed species, Shannon diversity index, and Faith's PD were calculated using the ASV table in QIIME2, and visualized as box plots. Beta-diversity analysis was performed to investigate the structural variation of microbial communities across samples using Jaccard metrics and Bray–Curtis metrics $via$ principal coordinate analysis (PCoA). The significance of differentiation of microbiota structure among groups was assessed by permutational multivariate analysis of variance (PERMANOVA) (48) and Permdisp (49) using QIIME2. Linear discriminant analysis effect size (LEfSe) was performed to detect differentially abundant taxa across groups using the default parameters (50). LDA was used to assess the effect size of each feature. The cut-off value of LDA score (log10) was 2. The correlation between saliva microbial communities and physiological influencing factors was performed using a linear mode of redundancy analysis (RDA) and correlation heatmap.

Statistical analysis was performed with SPSS (version 29.0; IBM). Continuous variables were reported as mean ± standard deviation for normally distributed variables, and as median (interquartile range) for non-normally distributed variables. Kruskal–Wallis test was performed to determine significant differences with regard to continuous

variables when multiple groups were compared. Categorical variables were reported as a proportion, and differences among groups were analyzed with Chi-squared test. Statistical significance was considered when two-sided $P < 0.05$.

## RESULTS

### Comparison of participants

A total of 112 children were recruited in the study. There was no significant difference in demographic characteristics among all the groups (Table 1). Children in the hypertrophy groups underwent overnight polysomnography, with χ test showing no significance in the prevalence of OSA and Kruskal–Wallis test revealing no significance in obstructive apnea–hypopnea index (OAHI) and lowest oxygen saturation (SaO$_2$) among the four hypertrophy groups ($P > 0.05$).

### Comparisons of salivary microbiota among children with different sites of upper airway obstruction

To identify the variation of microbiome in saliva from children with different sites of upper airway obstruction, comparisons were made among TH, AH, ATH, and control groups. Children diagnosed with Grade III tonsillar hypertrophy were selected representatively as the TH group here. A total of 9,170,314 raw reads were generated from 92 samples in this study, and 8,130,198 high-quality reads were retained after filtering and denoising. On average, 88,372 reads per sample were obtained for analysis. The average length of each read was 426 bp (range: 404–431 bp).

Compared with the AH, ATH, and control groups, the TH group had significantly higher alpha-diversity indices, including Chao1, Faith's pd, Shannon, and Observed species (Fig. 1A, $P < 0.05$, Kruskal–Wallis test), indicating that the microbial community in the saliva had a higher richness and was more evenly distributed in pediatric tonsillar hypertrophy patients.

Beta diversity among samples was visualized using principal coordinate analysis (PCoA) based on Bray–Curtis distance (Fig. 1B). PCoA revealed that the salivary microbiome of participants in the TH group had a slight separation from the participants in the other three groups (TH vs. AH: $P < 0.05$; TH vs. ATH: $P < 0.01$; TH vs. control: $P < 0.01$; PERMANOVA), and no obvious separation was found among AH, ATH, and control groups, indicating positive changes in the saliva microbial profile of pediatric tonsillar hypertrophy. Permutational multivariate analysis of dispersion (PERMDISP) was also performed to further test the homogeneity of within-group multivariate dispersions.

**TABLE 1** Basic demographic characteristics and polysomnograhy data of the participants[c]

| | TH group | | AH group | ATH group | Control group | P-value |
|---|---|---|---|---|---|---|
| | TH$_L$ ($n = 26$) | TH$_S$ ($n = 20$) | ($n = 21$) | ($n = 23$) | ($n = 22$) | |
| Age, years | 8.4 ± 2.5 | 8.4 ± 2.4 | 7.6 ± 2.8 | 7.2 ± 2.1 | 8.6 ± 2.1 | 0.107[a] |
| Sex | | | | | | |
| Male | 17 (65.4%) | 12 (60.0%) | 13 (61.9%) | 15 (65.2%) | 14(63.6%) | 0.995[b] |
| Female | 9 (34.6%) | 8 (40.0%) | 8 (38.1%) | 8 (34.8%) | 8(36.4%) | |
| BMI, kg/m$^2$ | 20.0 (14.9, 22.7) | 17.71 (14.7, 23.8) | 15.5 (14.4, 19.0) | 18.1 (14.7, 21.5) | 17.1 (15.2, 18.1) | 0.315[a] |
| OSA | | | | | | |
| Yes | 18 (69.2%) | 13 (65.0%) | 11 (52.4%) | 18 (78.3%) | n.d. | –[d] |
| No | 8 (30.8%) | 7 (35.0%) | 10 (47.6%) | 5 (21.7%) | n.d. | |
| OAHI, events/h | 1.6 (0.5, 2.9) | 1.4 (0.6, 2.6) | 1.1 (0.3, 3.8) | 3.0 (1.1, 8.9) | n.d. | – |
| Lowest-SaO$_2$, % | 93.5 (92, 96) | 93 (90, 95) | 93.0 (90.5, 97.5) | 90.0 (87.0, 94.0) | n.d. | – |

[a]Kruskal-Wallis test.
[b]Chi-square test.
[c]TH, tonsillar hypertrophy; TH$_L$, larger tonsillar hypertrophy; TH$_S$, smaller tonsillar hypertrophy; AH, adenoid hypertrophy; ATH, adenotonsillar hypertrophy; BMI, body mass index; OAHI, obstructive apnea hypopnea index; SaO$_2$, oxygen saturation; n.d., not determined.
[d]"-" Indicates that the p-value of this corresponding indicator cannot be calculated among the four groups.

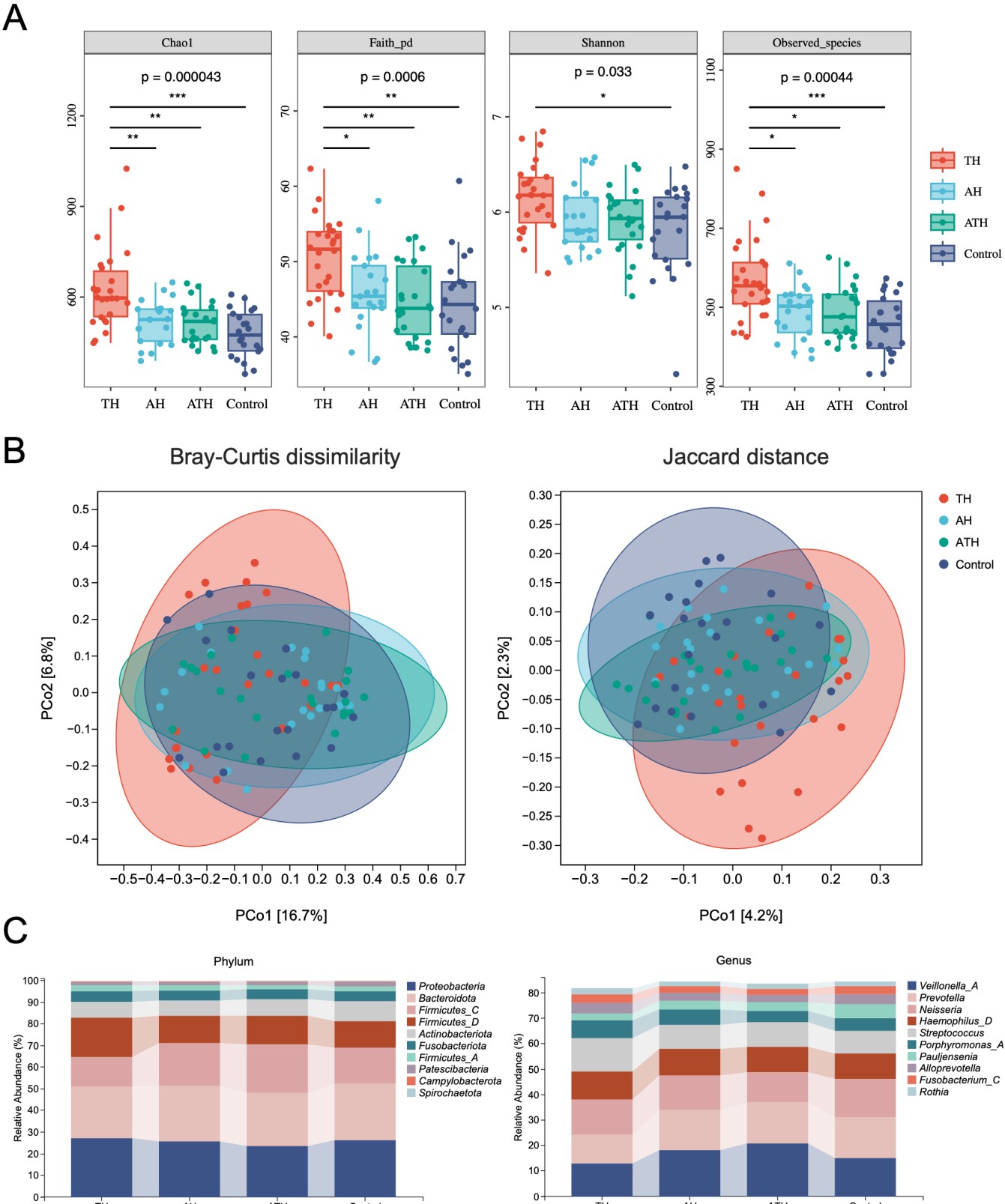

**FIG 1** Comparison of saliva microbiota diversity and composition among TH, AH, ATH and control groups. (A) Alpha diversity of each group measured with Chao1, Faith's pd, Shannon and Observed species indices. From bottom to top, each box plot represents the minimum, first quartile, median, third quartile and maximum values. Statistical analysis was performed using Kruskal–Wallis and Dunn's post hoc tests (*P < 0.05, **P < 0.01, and ***P < 0.001).

Fig 1 (Continued)

(B) Beta-diversity-based principal coordinate analysis (PCoA) plots of each group measured with Bray–Curtis dissimilarity (abundance‑based) and Jaccard distance (composition‑based). Each point represents the sample. The numbers in square brackets represent the percentage of the total variance explained by the principal coordinates. (C) The top 10 microbial composition at the phylum and genus level in each group. AH, adenoid hypertrophy group; TH, tonsillar hypertrophy group; ATH, adenotonsillar hypertrophy group.

The result indicated that the intra-group variances were not significantly different ($P >$ 0.05, PERMDISP).

Microbial composition in the TH, AH, ATH, and control groups at the phylum and genus levels are displayed in Fig. 1C. Among TH children, the top five genus-level microorganisms were *Neisseria* (14.03%), *Streptococcus* (13.05%), *Veillonella_A* (12.88%), *Prevotella* (11.25%), and *Haemophilus_D* (11.06%). Comparing the TH group with AH, ATH, and control groups, the relative abundances of phyla *Proteobacteria* and *Firmicutes_D* were the highest in the TH group than in other three groups. At the genus level, the relative abundances of *Veillonella_A*, *Prevotella*, *Pauljensenia* were the lowest in the TH group and that of *Neisseria*, *Haemophilus_D*, *Streptococcus*, *Porphyromonas_A*, *Alloprevotella*, and *Fusobacterium_C* were the highest in TH group than in other three groups.

## Taxa alteration in children with tonsillar hypertrophy

To further identify vital taxonomic differences between tonsillar hypertrophy group and other groups, we conducted LEfSe analysis and found significant abundance differences in salivary microbiota among the four groups (Fig. 2, LDA scores (log10) >2, $P < 0.05$).

LEfSe analysis revealed that 47 taxa were significantly most abundant in the saliva samples from the pediatric patients with tonsillar hypertrophy, whereas six taxa were significantly most abundant from the adenotonsillar hypertrophy children, and four taxa were significantly most abundant from healthy controls. There were no taxa significantly abundant from adenoid hypertrophy children (Fig. 2A).

A cladogram shown in Fig. 2B represented the connection between the significantly different taxa at different taxonomic levels. For example, *Gemella* (genus) is under *Gemellaceae* (family), which is under *Staphylococcales* (order). A clade is a branch of organisms under a common ancestor. The significantly different taxa were shown in a tree-like structure. Moreover, LDA score demonstrated that these differentially abundant taxa can be considered as potential biomarkers (LDA score > 2, $P < 0.05$).

Selected genus with significant differences between the TH group and other groups were summarized using box plots (Fig. 3). Compared with children with adenoid hypertrophy, adenotonsillar hypertrophy, and healthy controls, salivary microbiota from children with tonsillar hypertrophy were mainly characterized by a significantly higher abundance of genera, such as *Gemella* ($P < 0.001$), *Haemophilus* ($P < 0.01$), *Parvimonas* ($P < 0.01$), *Dialister* ($P < 0.05$), *Filifactor* ($P < 0.01$), *Bacillus* ($P < 0.001$), *Lactobacillus* ($P < 0.05$), *Bifidobacterium* ($P < 0.001$), and *Treponema* ($P < 0.05$). Meanwhile, genera, such as *Veillonella*, *Prevotella*, *Lancefieldella*, and *Anaeroglobus* were least abundant in tonsillar hypertrophy children ($P < 0.05$).

## Comparisons of salivary microbiota between children with different degrees of tonsillar hypertrophy

To better understand the tonsillar hypertrophy salivary microbiome characteristics, we performed additional analyses between children with Grade II (TH$_S$) and Grade III (TH$_L$) tonsillar hypertrophy. Comparing the TH$_S$ group with the TH$_L$ group, no significant differences were shown in alpha diversity as estimated by Faith's pd, Shannon, and Observed species (Fig. 4A, Mann–Whitney U-test, $P > 0.05$). PCoA revealed that the salivary microbiota of children from the TH$_S$ and TH$_L$ groups had no obvious separation based on the Bray–Curtis dissimilarities ($P = 0.323$, PERMANOVA) and Jaccard distances

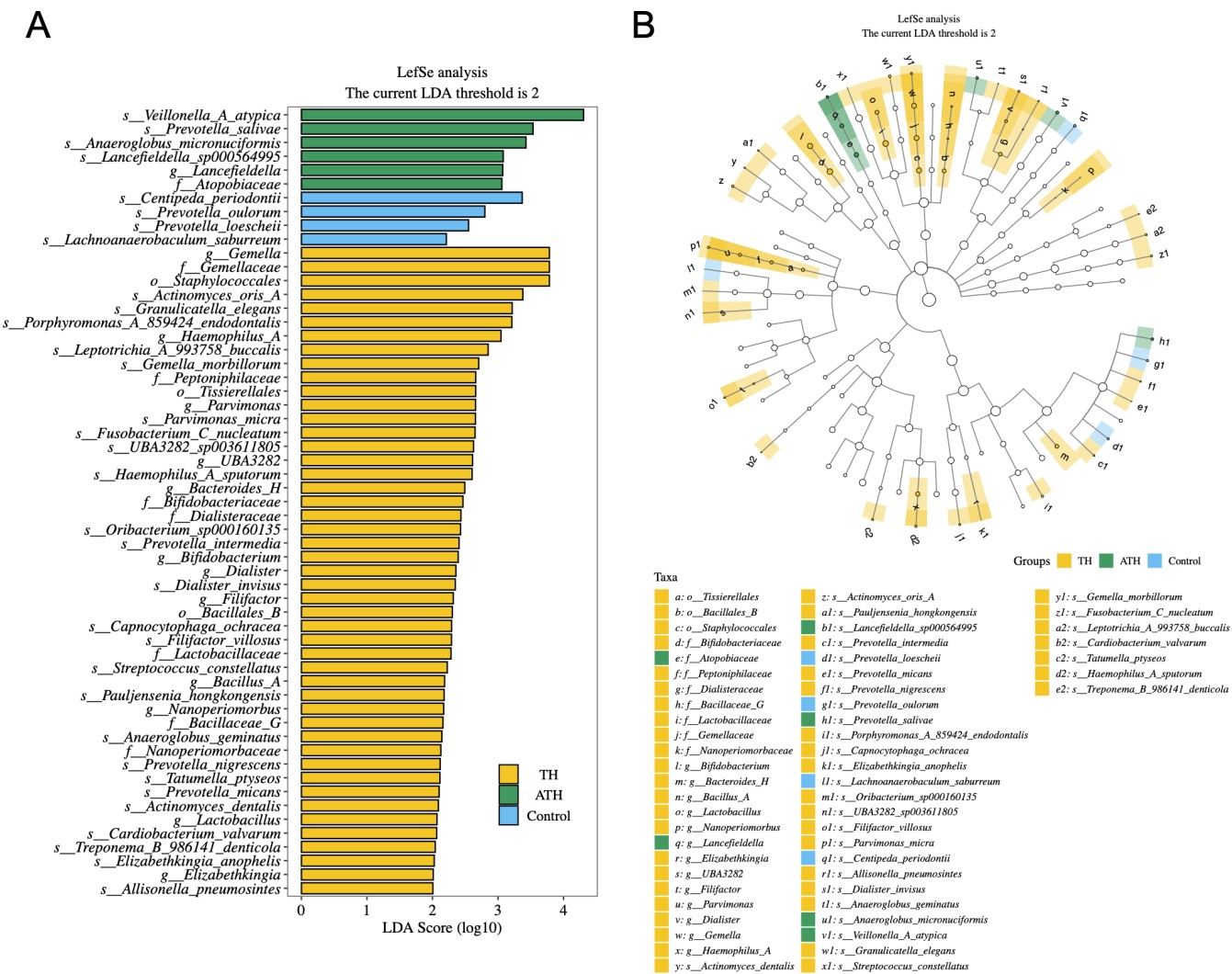

**FIG 2** Linear discriminant analysis (LDA) effect size (LEfSe) analysis of microbiota composition. (A) Histogram of LDA scores showing significant taxonomic differences among the TH (yellow), ATH (green), and control groups (blue). The longer the length, the more significant the difference in the taxa. Only the LDA scores >2 are listed. (B) Cladogram plotted from LEfSe analysis showing that the colored nodes from inner circle to outer circle represented the hierarchical relationship of all taxa from the phylum to the species level. Taxa enriched in the TH, ATH, and control groups were shown in yellow, green, and blue. Taxa with nonsignificant changes were represented by empty circles. The diameter of each small circle represented the taxa abundance. Only the LDA scores >2 are listed. TH, tonsillar hypertrophy group; AH, adenoid hypertrophy group; ATH, adenotonsillar hypertrophy group.

($P$ = 0.707, PERMANOVA), indicating that there was basically no difference in the salivary microbial diversity between children with tonsillar hypertrophy.

## Correlations between salivary microbiome and physiological influencing factors

For TH, AH, and ATH groups, 21 children in each group had both results of PSG and CBC. Figure 5A shows the CBC parameters exhibiting differences among the three groups of children, and there were no statistically significant differences in other tested hematological outcomes. The mean corpuscular hemoglobin concentration (MCHC) and red cell distribution width (RDW) of children in TH group were lower than those in AH and ATH groups, whereas platelet (PLT) and plateletcrit (PCT) were higher compared with other groups. An effect size redundancy analysis (RDA) was conducted to determine which factors explained the variation in microbial composition across samples (Fig. 5B). A total of 25 non-redundant factors were included in the RDA, including 23 CBC parameters and

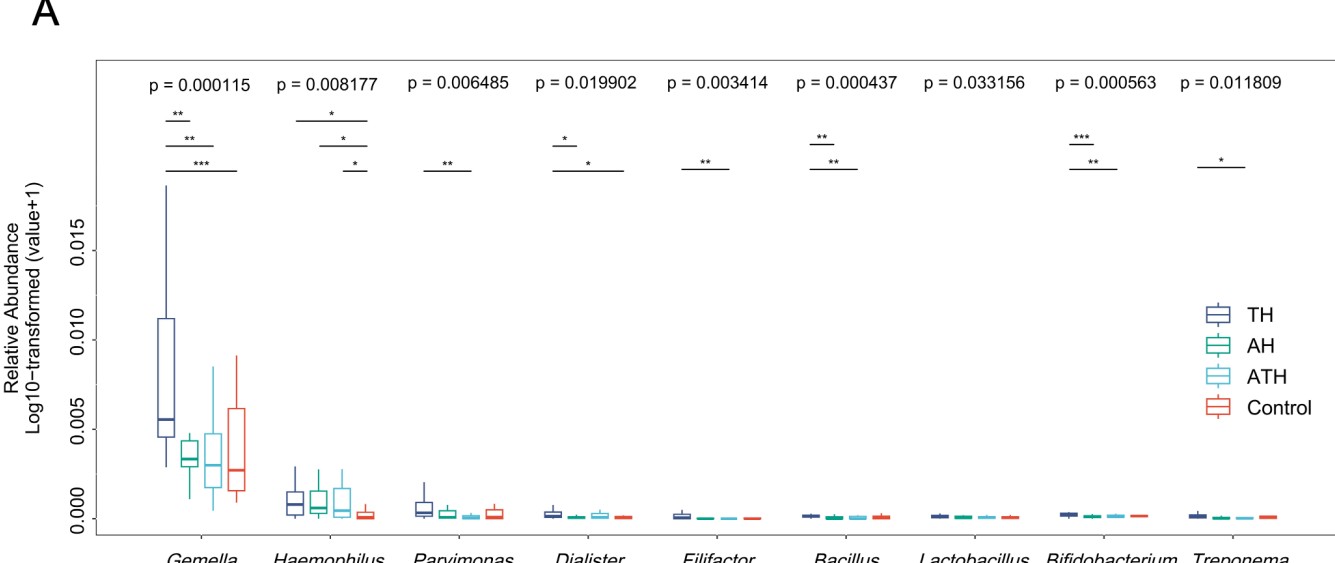

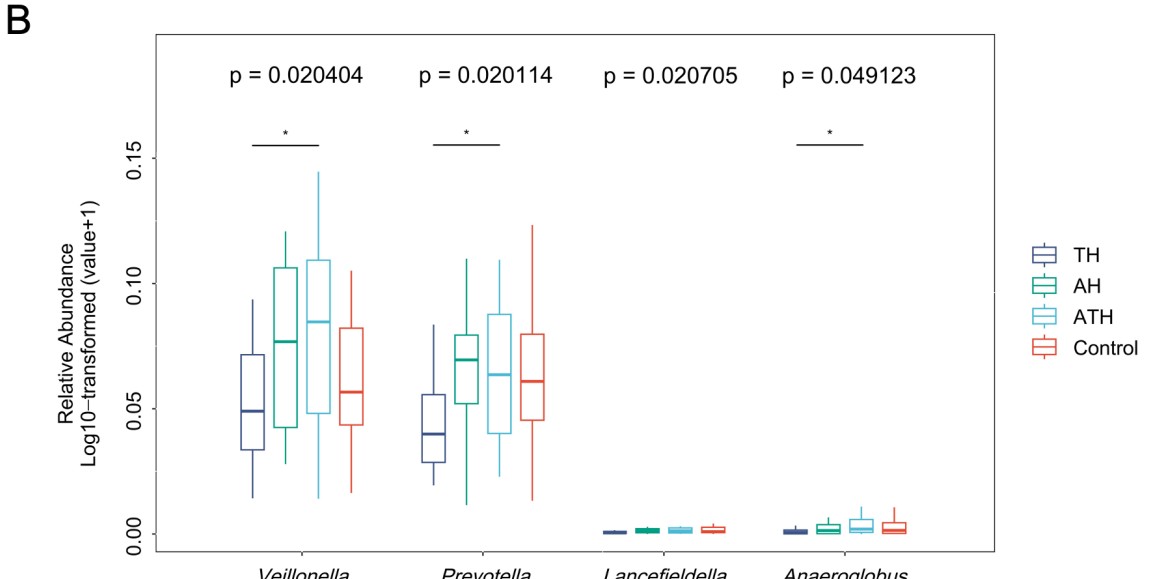

**FIG 3** Relative abundance of genus enriched and least enriched in the TH group verified by linear discriminant analysis effect size (LEfSe) analysis. (A) Boxplots showing log10-transformed (value +1) relative abundances of genera with the most significant enrichments in the TH group than the other three groups. (B) Boxplots showing log10-transformed (value +1) relative abundances of genera with the least enrichments in TH group than the other three groups. Boxes represent the interquartile ranges, and lines inside the boxes denote medians. P values by Kruskal–Wallis test are indicated. *$P < 0.05$, **$P < 0.01$, ***$P < 0.001$. TH, tonsillar hypertrophy group; AH, adenoid hypertrophy group; ATH, adenotonsillar hypertrophy group.

2 PSG indices. The factor that explained the most variation was the platelet distribution width (PDW). In addition, PSG indices, OAHI, and lowest $SaO_2$, correlated closely with MCHC, RDW, RBC, Hb, and WBC, which may be associated with the occurrence of hypoxemia and inflammation. Further, we investigated the correlation of physiological influencing factors with the abundance of the microbiota in TH children using correlation heatmap (Fig. 5C). The genus *Treponema* was highly positively correlated with mean platelet volume (MPV), platelet distribution width (RDW), and large platelet ratio (LPR), and *Filifactor* was positively correlated to RDW as well, whereas *Haemophilus* was significantly negatively associated with lowest $SaO_2$.

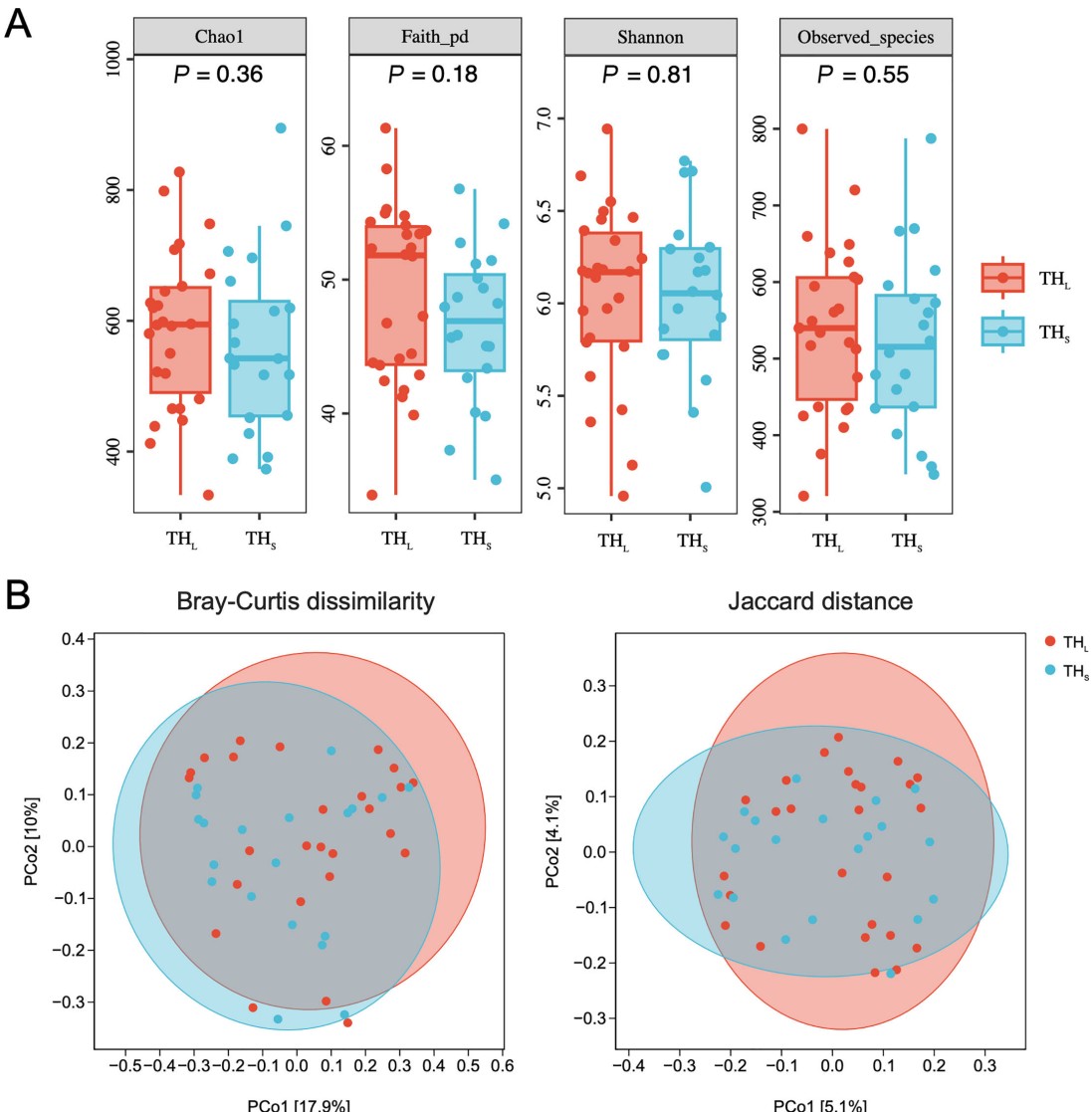

**FIG 4** Comparison of salivary microbial diversity between the TH$_S$ and TH$_L$ groups. (A) Alpha diversity of each group measured with Chao1, Faith's pd, Shannon, and Observed species indices ($P < 0.05$, Mann–Whitney U-test). (B) Beta-diversity-based principal coordinate analysis (PCoA) plots of each group measured with Bray–Curtis dissimilarity (abundance - based) and Jaccard distance (composition - based). Each point represents the sample. The numbers in square brackets represent the percentage of the total variance explained by the principal coordinates. TH$_L$, larger tonsillar hypertrophy group; TH$_S$, smaller tonsillar hypertrophy group.

## DISCUSSION

In our previous study, we discovered that children with isolated adenoid hypertrophy and adenotonsillar hypertrophy shared similar craniofacial morphology features, whereas children with isolated tonsillar hypertrophy exhibited quite opposite characteristics (11). Here, we have also made a captivating finding that the salivary microbial profile in tonsillar hypertrophy children is unique, whereas the microbial diversity in adenoid hypertrophy and adenotonsillar hypertrophy children remains quite similar. To the best of our knowledge, this is the first study using high-throughput analysis to explore the salivary microbiome of children with different sites of upper airway obstruction, including tonsillar hypertrophy, adenoid hypertrophy, and adenotonsillar hypertrophy in pediatric Chinese population.

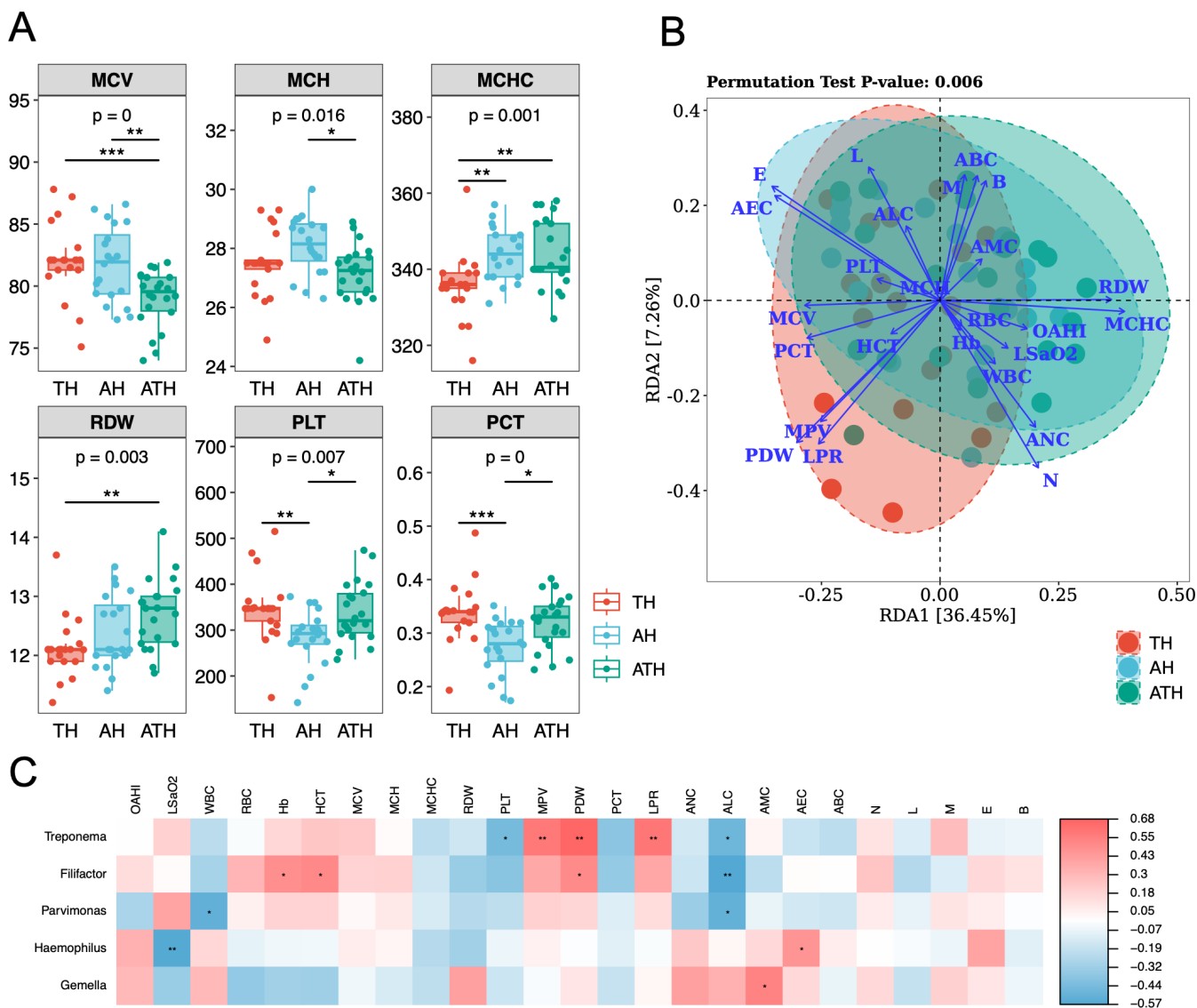

**FIG 5** Correlations between salivary microbiome and physiological influencing factors. (A) Box plots illustrating the physiological parameters that differ among the TH, AH, and ATH groups. Statistical analysis was performed using Kruskal–Wallis and Dunn's *post hoc* tests (\*P < 0.05; \*\*P < 0.01, and \*\*\*P < 0.001). (B) Redundancy analysis (RDA) plot depicting the relationship between saliva microbiome and physiological influencing factors explained by CBC and PSG results. Blue arrows represent different influencing factors, with the angle between them indicating the magnitude of their correlation. An acute angle indicates a positive correlation, a right angle indicates no correlation, and an obtuse angle indicates a negative correlation. The longer the ray, the greater the impact of the factor on microbial composition/functionality. *P*-value was obtained from the permutation test. (C) The heatmap of correlation between physiological influencing factors and each genus. Positive and negative correlations are represented by blue or red, respectively. The statistically significant correlation between two variables were tested by Pearson correlation coefficient, and annotated as \*P < 0.05, \*\*P < 0.01, \*\*\*P < 0.001. OAHI, obstructive apnea hypopnea index, LSaO$_2$, lowest oxygen saturation; WBC, white blood cell; RBC, red blood cell; Hb, hemoglobin; HCT, hematocrit; MCV, mean corpuscular volume; MCH, mean corpuscular hemoglobin; MCHC, mean corpuscular hemoglobin concentration; RDW, red cell distribution width; PLT, platelets; MPV, mean platelet volume; PDW, platelet distribution width; PCT, plateletcrit, LPR, large platelet ratio; N, neutrophils; L, lymphocytes; M, monocytes; E, eosinophils; B, basophils; ANC, absolute neutrophil count; ALC, absolute lymphocyte count; AMC, absolute monocyte count; AEC, absolute eosinophil count; ABC, absolute basophil count.

## Salivary microbial diversity in pediatric tonsillar hypertrophy

In the present study, we concluded that the salivary microbiome of TH children had a significantly higher alpha diversity than the AH, ATH, and control groups and illuminated distinct differences in the beta diversity of the salivary microbiome. These results are inconsistent with the research conducted by Xu et al., which found no statistical

differences in alpha or beta diversity between the control and tonsillar hypertrophy groups with 14 TH children and 12 healthy children (29). The difference may be explained by two possible reasons. First, according to Micropower package (36), we suggested at least 20 subjects per group to adequately demonstrate statistical significance in microbiome studies (25). Second, their fasting period requirement (2 h) was shorter than ours (6 h). It was indicated that a fast of at least 6 h was believed to effectively prevent the contamination of other DNA sequences, PCR inhibitors, and other substances (51), and studies on gut microbiome found that different fasting protocols could significantly impact bacterial populations, as well (52). In alignment with our result, previous studies have also reported *Streptococcus*, *Haemophilus, Veillonella*, and *Prevotella* as the most common bacteria found in enlarged tonsils (21, 53), and the tonsillar crypts of children with tonsillar hypertrophy were dominated by the following genera: *Streptococcus* (21.5%), *Neisseria* (13.5%), *Prevotella* (12.0%), and *Haemophilus* (10.2%) (23). Meanwhile, our data showed that the diversity of salivary microbiome in children with Grade II and Grade III tonsillar hypertrophy was largely similar, suggesting that it is the site of obstruction predominantly influencing the salivary microbiota, rather than the severity of hypertrophy. Although obstructive apnea–hypopnea index (OAHI) was slightly higher in the ATH group, the inter-group differences were not statistically significant. Thus, OAHI was a balanced indicator in this study. The existence of a distinctive microbiome profile in saliva from children with tonsillar hypertrophy and adenoid hypertrophy/adenotonsillar hypertrophy may indicate that different sites of upper airway obstruction may simultaneously lead to changes in craniofacial morphology and microbial profile.

## Characteristic microbiota in pediatric tonsillar hypertrophy

As the microbial composition was proven to interact with the characteristic phenotype of tonsillar disease (54), the characteristic bacteria of children with tonsillar hypertrophy are another focus of our study. *Gemella*, a predominant genus of the mucosal epithelium (55), had a significantly higher abundance in the TH group compared with the AH, ATH, and control groups in our study. When comparing tonsillar microbiota from patients with chronic tonsillitis and tonsillar hypertrophy, Wu et al. also noted a higher abundance of *Gemella* in one tonsillar microbial type, which was mainly identified in core tonsils from TH patients (18). Ko et al. found that *Gemella* might play a role in the hypoxia state since the abundance of *Gemella* was decreased in OSA patients after CPAP treatment (56). The function of *Gemella* was considered to correspond to Th2-mediated anti-inflammatory reaction described by Wang et al. (57). Thorsen et al. further reported that a higher abundance of *Gemella* was associated with higher CCL2 and CCL17, the Th2 cell-specific chemokines, and could increase the risk of developing childhood asthma (58). Notably, asthma-related symptoms might be exacerbated by tonsillar hypertrophy (59). Meanwhile, *Gemella* was identified to be associated with the occurrence of dental caries in deciduous teeth (60), related to allergy development particularly at 7 years of age (61), which coincided with the mean age of the participants in our study, and the abundance was even higher in patients with oral cancer than in healthy people (62). Whether the enrichment of *Gemella* persists in other age groups still requires further investigation. Our results also revealed a positive correlation between the abundance of *Gemella* and absolute monocyte count, supporting the interaction between *Gemella* and the immune response. We therefore indicate that *Gemella* is most closely related to tonsillar hypertrophy with profound underlying mechanisms in immunomodulation.

Consistent with the results of other researchers (63), we also found that genera *Parvimonas*, *Dialister*, *Bacillus*, *Filifactor*, and *Lactobacillus* exhibited significantly higher abundance in TH children, all of which belong to phylum *Firmicutes*, suggesting that phylum *Firmicutes* forms the majority of the characteristic microbiota in the saliva of tonsillar hypertrophy children. Alongside *Gemella*, genera *Parvimonas*, *Dialister* , and *Lactobacillus* also contribute to immune regulation. Particularly, Zhao et al. identified that *Parvimonas* promotes colorectal cancer development via enhancing Th17-mediated immune response (64), and *Parvimonas* strains were assumed to be able to facilitate a

strong immune response of the sepsis marker calprotectin (65). It has also been reported that *Dialister* could exert an immunomodulatory effect by producing the short-chain fatty acid (SCFA) pentanoate (66). Rastogi et al. summarized the evidence of the benefit of *Lactobacillus* in acting as powerful immune cell controllers and exhibiting a regulatory role pertaining to different organs from animal and clinical trials (67). These reports coupled with our findings suggest that the group composed of *Gemella*, *Parvimonas*, *Dialister*, and *Lactobacillus* could involve in the immune response in tonsillar hypertrophy and reflect an active state of immune regulation, but the specific mechanism and its potential use as an early diagnostic marker in tonsillar hypertrophy deserves further investigation.

In our study, *Filifactor* and *Treponema* had the highest relative abundance in TH children and positively related to PDW and MPV, markers of platelet activation (68). *Filifactor* was considered to contribute in tonsillar hypertrophy (69), and its species has been identified as one of the most commonly associated pathogens with platelet activation and aggregation (70), all of which are also corroborated in our study. Recent studies have revealed that specific species of *Treponema* can be involved in IgA nephropathy, known as one of the tonsil-related diseases (71), and a high relative abundance of *Treponema* was noted in the oral cavity of patients with halitosis, a common symptom of tonsillar hypertrophy (72), and in patients with recurrent tonsillitis (73). These results substantiate the close association of *Filifactor* and *Treponema* with the unhealthy state of tonsils. Previous studies have found that PDW was higher in severe OSA and was correlated with different parameters of breathing function during sleep (74). MPV was significantly higher in pediatric patients with sleep-disordered breathing (75). PDW and MPV levels increase in hypoxia conditions *via* elevated IL-6 levels that can lead to platelet activation (76, 77). Thus, we speculated that the high abundance of *Filifactor* and *Treponema* may contribute to platelet activation caused by hypoxia resulting from tonsillar hypertrophy, which necessitates further exploration.

We also noted the discrepancies in characteristic genera discovered in tonsillitis in previous studies and those identified in our work. *Veillonella* and *Prevotella*, which were frequently detected in tonsillitis (23, 78) and have been strongly linked to pro-inflammation abilities, had conversely significantly lower abundance in the TH group than in other groups. *Veillonella* was associated with poor oral health, leading to various oral infectious diseases, such as periodontitis (79), and found to be associated with bleeding beyond 24 h after tonsillectomy (80). Abe et al. indicated that *Veillonella* positively correlated with the levels of pro-inflammatory cytokines, such as IL-1β, IL-6, IL-8, and IL-12p70 in the saliva of patients with autoimmune hepatitis (81). Meanwhile, emerging studies in humans have associated increased abundance in specific strains of *Prevotella* with localized and systemic inflammatory disorders, including periodontitis (82), gingivitis (83), and metabolic disorders (84). *Prevotella* was also found to be positively correlated with low-density lipoprotein (LDL), supporting its role in pro-inflammation (85). Moreover, *Haemophilus influenzae* has been implicated as an etiological role in the pathogenesis of tonsillitis in previous studies mostly using tonsil tissue or oropharyngeal swab samples based on bacterial culture (86–89), and *Haemophilus* was also enriched in the TH group in our study with saliva samples, suggesting that the microbial relationship between tonsillar hypertrophy and tonsillitis is more intricate and not completely distinct.

## Limitations

Based on the collective results, we have enhanced the comprehension of the salivary microbiome profile of pediatric tonsillar hypertrophy. However, some limitations should be noted. First, we did not verify the causal relationship between characteristic salivary microbiota and tonsillar hypertrophy, and the specific immunomodulatory mechanisms of the microbiome on tonsillar hypertrophy have not been further investigated. Second, children in the control group did not undergo otolaryngology examinations consistent with children in hypertrophy groups, such as fibro-laryngoscopic examination for

adenoid. Third, to exclude the effects of inflammation and antibiotic use, we only enrolled children who reported no history of acute tonsillitis or recent clinical events of inflammation, but mild inflammation may not necessarily exhibit obvious symptoms; thus, the hypertrophy groups in this study may not purely represent hypertrophy without any accompanying inflammation. Fourth, whole genome sequencing can offer higher species-level accuracy and provide more metabolic features compared with 16S rRNA sequencing and could be utilized in future studies to further our understanding of the functional capacities of microbial communities. Finally, our study did not conduct *in vivo* studies using animal models to validate the impact of upper airway obstruction on the salivary microbiome and how they interact. Future animal experiments are needed to elucidate more definitive mechanisms.

## Conclusion

The analysis of the microbiome of saliva samples from children with tonsillar hypertrophy and/or adenoid hypertrophy revealed that the site of upper airway obstruction indeed exerts an impact on the salivary microbiome as we expected. Our study reinforces the distinctiveness of tonsillar hypertrophy in children and links them to the characteristic genera comprising *Gemella*, *Parvimonas*, *Dialister*, and *Lactobacillus* with immune regulatory capabilities. Future studies could be directed toward the candidate microbial genera and identifying the underlying mechanisms.

## ACKNOWLEDGMENTS

We are grateful for the support from all the participated children and their families. We would also like to extend our gratitude to Zhifei Xu and Li Zheng from the Sleep Center of Beijing Children's Hospital for the assistance in sample collection.

This study was funded by the National Natural Science Foundation of China (82170102).

## AUTHOR AFFILIATIONS

[1]Department of Orthodontics, Peking University School and Hospital of Stomatology, Beijing, China
[2]Department of Stomatology, Beijing Friendship Hospital, Capital Medical University, Beijing, China
[3]Department of Otolaryngology, Head and Neck Surgery, Beijing Children's Hospital, Capital Medical University, National Center for Children's Health, Beijing, China

## AUTHOR ORCIDs

Ying Xu http://orcid.org/0009-0004-2146-3992
Jie Zhang http://orcid.org/0000-0002-1621-7779
Xuemei Gao http://orcid.org/0000-0001-5690-9385

## FUNDING

| Funder | Grant(s) | Author(s) |
| --- | --- | --- |
| MOST | National Natural Science Foundation of China (NSFC) | 82170102 | Xuemei Gao |

## AUTHOR CONTRIBUTIONS

Jie Zhang, Conceptualization, Project administration, Resources.

## DATA AVAILABILITY

The data presented in the study are deposited in the Sequence Read Archive (https://www.ncbi.nlm.nih.gov/), accession number PRJNA1095809. The STORMS checklist for this article is available from Zenodo (10.5281/zenodo.12770455).

## ETHICS APPROVAL

The study was performed in accordance with the Declaration of Helsinki. It was reviewed and approved by Peking University School and Hospital of Stomatology Ethics Committee (Issuing number: PKUSSIRB-201947093) and was registered in the Chinese Clinical Trial Registry (ChiCTR2000041263). Informed consent was obtained from the parents or legal guardians of all participants prior to the initiation of the study. Additionally, child assent was obtained from participants that were considered to be capable of providing assent, considering their age, maturity, and psychological state.

## ADDITIONAL FILES

The following material is available online.

### Supplemental Material

**Table S1 (mSystems00968-24-s0001.docx).** The method of determining sample size.

### Open Peer Review

**PEER REVIEW HISTORY (review-history.pdf).** An accounting of the reviewer comments and feedback.

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
