## [Reviewer comments · mSystems]

Differences in salivary microbiome among children with tonsillar hypertrophy and/or adenoid hypertrophy

Ying Xu, Min Yu, Xin Huang, Guixiang Wang, Hua Wang, Fengzhen Zhang, Jie Zhang, and Xuemei Gao

Corresponding Author(s): Xuemei Gao, Peking University School and Hospital of Stomatology

Review Timeline:

Submission Date:

July 19, 2024

Accepted:

August 23, 2024

Editor: Neha Sachdeva

Reviewer(s): The reviewers have opted to remain anonymous.

Transaction Report:

DOI: <https://doi.org/10.1128/mSystems.00968-24>

Re: mSystems00968-24 (Differences in salivary microbiome among children with tonsillar hypertrophy and/or adenoid hypertrophy)

Dear Prof. Xuemei Gao:

Your manuscript has been accepted, and I am forwarding it to the ASM production staff for publication. Your paper will first be checked to make sure all elements meet the technical requirements. ASM staff will contact you if anything needs to be revised before copyediting and production can begin. Otherwise, you will be notified when your proofs are ready to be viewed.

Sincerely,
Neha Sachdeva
Editor
mSystems

Reviewer #1 (Comments for the Author):

The reviewer's comments were determined to have been adequately addressed, properly handled, and corrected and are therefore deemed acceptable.